# Flexible nitrogen utilisation by the metabolic generalist pathogen *Mycobacterium tuberculosis*

Aleksandra Agapova[1], Agnese Serafini[1], Michael Petridis[1], Debbie M Hunt[1], Acely Garza-Garcia[1], Charles D Sohaskey[2], Luiz Pedro Sório de Carvalho[1]*

[1]Mycobacterial Metabolism and Antibiotic Research Laboratory, The Francis Crick Institute, London, United Kingdom; [2]Department of Veterans Affairs Medical Center, Long Beach, United States

**Abstract** Bacterial metabolism is fundamental to survival and pathogenesis. We explore how *Mycobacterium tuberculosis* utilises amino acids as nitrogen sources, using a combination of bacterial physiology and stable isotope tracing coupled to mass spectrometry metabolomics methods. Our results define core properties of the nitrogen metabolic network from *M. tuberculosis*, such as: (i) the lack of homeostatic control of certain amino acid pool sizes; (ii) similar rates of utilisation of different amino acids as sole nitrogen sources; (iii) improved nitrogen utilisation from amino acids compared to ammonium; and (iv) co-metabolism of nitrogen sources. Finally, we discover that alanine dehydrogenase is involved in ammonium assimilation in *M. tuberculosis*, in addition to its essential role in alanine utilisation as a nitrogen source. This study represents the first in-depth analysis of nitrogen source utilisation by *M. tuberculosis* and reveals a flexible metabolic network with characteristics that are likely a product of evolution in the human host.

DOI: https://doi.org/10.7554/eLife.41129.001

*For correspondence:
luiz.carvalho@crick.ac.uk

Competing interests: The authors declare that no competing interests exist.

## Introduction

Human tuberculosis, caused by the bacillus *Mycobacterium tuberculosis*, is now the greatest cause of death by a single infectious agent, surpassing deaths caused by HIV/AIDS (*WHO, 2017*). Tuberculosis is a complex and unique disease, whereby *M. tuberculosis* evades eradication by the immune system and often by chemotherapy with antibiotics. As with other bacterial diseases, tuberculosis is increasingly drug resistant, with strains resistant to two front-line drugs, isoniazid and rifampicin (multi-drug resistant), and additional resistance to an injectable antibiotic and a quinolone (extensive-drug resistant) killing now over 500 thousand patients out of a total of 1,800,000 deaths (*WHO, 2017*). Although novel first-in-class antitubercular agents have been discovered in the last 20 years (*Tiberi et al., 2018*), resistance to these agents and important side effects might preclude their widespread utilization and hence the reversal of the epidemic.

We and others believe that failures in drug discovery programmes aimed at finding transformative antitubercular agents are in large part caused by our incomplete understanding of bacterial phenotypic diversity in the host (*Barry et al., 2009*). Bacterial metabolic flexibility is thought to be essential for growth and survival in a variety of niches, where low pH, low oxygen tension, presence of reactive oxygen and nitrogen species, and scarcity of nutrients are commonly found. Although we are aware and partially understand some of the effects of low oxygen and mild acidic pH on *M. tuberculosis* growth and metabolism (*Eoh and Rhee, 2013*; *Eoh et al., 2017*), we clearly do not know the full complement of conditions and niches occupied by *M. tuberculosis*. In addition, we do

**eLife digest** Tuberculosis is an infectious disease caused by a bacterium called *Mycobacterium tuberculosis*. It is currently the leading cause of death by a single microbe worldwide, claiming the lives of 1.5 million people annually. The disease is difficult to cure, as many strains of the bacterium have developed resistance to the main drugs used to treat the infection. This leaves physicians with few options to treat tuberculosis and control its spread. The spread of these drug-resistant strains is a major global public health problem.

New strategies that do not lead to drug resistance are needed. One possibility would be to starve the bacterium. Like all living things, *M. tuberculosis* must eat to survive and spread. Right now, scientists do not know much about how this microbe eats. However, they do know that it needs nitrogen – an essential part of DNA, RNA, and proteins – to survive. Most bacteria like to consume ammonium as their main nitrogen source, but they may also use select amino acids as a nitrogen source.

Now, Agapova et al. show that *M. tuberculosis* is not a picky eater. In the experiments, the bacteria were fed different nitrogen sources. Then, they tracked how well the bacteria grew. The experiments showed that *M. tuberculosis* happily eats many different amino acids and may use more than one as a nitrogen source at a time. It does not tightly control its stockpile of nitrogen sources the way other bacteria do, or use ammonium very efficiently.

This suggests that *M. tuberculosis* has evolved to be very flexible in its dietary habits, which may explain why these bacteria can thrive in the varied environments within the human body. Knowing exactly how *M. tuberculosis* acquires and uses nitrogen may help scientists design ways to thwart the process and starve the bacteria.

DOI: https://doi.org/10.7554/eLife.41129.002

not understand more fundamental aspects of *M. tuberculosis* nutrition and metabolism even in pure cultures devoid of host cells.

Despite all research carried out with *M. tuberculosis* in the last decades, we still lack fundamental understanding on how its unique metabolic network promotes survival in the human host, pathogenesis, and long-term persistence (*Rhee et al., 2011*; *Ehrt et al., 2018*). Most current knowledge of host-relevant *M. tuberculosis* metabolism spans central carbon metabolism (e.g. glycolysis, gluconeogenesis, glyoxylate bypass) (recently reviewed in *Ehrt et al., 2018*). In contrast, very little is known about nitrogen metabolism, in particular, we do not understand what are the essential features of nitrogen metabolism in *M. tuberculosis* (*Gouzy et al., 2014a*). For example, while we understand how post-translational regulation of nitrogen metabolism operates in mycobacteria (*Carroll et al., 2008*; *Cowley et al., 2004*; *Nott et al., 2009*; *O'Hare et al., 2008*; *Pashley et al., 2006*; *Read et al., 2007*; *Rieck et al., 2017*; *Parish and Stoker, 2000*), transcriptional regulation of nitrogen metabolism in *M. tuberculosis* is largely unknown. *M. tuberculosis* lacks homologues for nearly all known bacterial transcriptional factors involved in nitrogen metabolism in other bacteria *Figure 1—figure supplement 1*, and the transcriptional factor *GlnR* does not perform canonical functions (*Williams et al., 2015*). Instead, it appears that *GlnR* regulates ammonia and nitrate uptake in *M. tuberculosis* (*Williams et al., 2015*). Simple comparison of growth kinetics in identical culture medium reveals that *M. smegmatis* and *M. tuberculosis* growth differs significantly, not only due to inherent growth rate differences, but also lag phase and final biomass achieved *Figure 1—figure supplement 2*, points to species-specific variations in nitrogen metabolism. Importantly, the vast majority of studies to date focused exclusively on either ammonium ($NH_4^+$) as the sole physiologically relevant nitrogen source (*Williams et al., 2015*; *Petridis et al., 2015*) or employed surrogate fast-growing species such a *M. smegmatis* instead of *M. tuberculosis* (*Petridis et al., 2015*). For example, *GlnR* of *M. smegmatis* is evolutionarily closer to the homologues present in *Nocardia farcina* and *Rhodococcus sp.* RHA1 than those found in *M. tuberculosis*, *M. bovis* and *M. avium* (*Amon et al., 2009*), which could indicate different functions of *GlnR* within the *Mycobacterium* genus. This discrepancy indicates that *M. smegmatis* cannot be used as a model of *M. tuberculosis* with regards to nitrogen metabolism.

A number of studies were published at the time the now common culture media for in vitro growth of *M. tuberculosis* were developed (*Proskauer and Beck, 1894*). These studies provided rigorous and systematic analysis of the effect of amino acids as growth promoters for *M. tuberculosis*. In spite of this, four main issues preclude a deeper interpretation of the results obtained. First, cultures were not pre-adapted in the amino acids tested as sole nitrogen sources, leading to results that are likely partially distorted due to the nitrogen sources in the prior culture media. Second, cultures often contain multiple nitrogen sources, which complicate the analysis. Third, only recently nitrogen tracing was made possible by the use of modern high-resolution mass spectrometers and the use of labelled and position-specific labelled nitrogen sources. Finally, in a number of studies, qualitative results were reported instead of doubling times or growth rates, precluding direct comparisons between studies, conditions and/or species. In 2013–2014, two key studies unveiled an important aspect of host-relevant nitrogen metabolism in *M. tuberculosis*, namely that host amino acids such as L-aspartate (Asp) and L-asparagine (Asn) are important sources of nitrogen during infection (*Gouzy et al., 2014b*; *Gouzy et al., 2013*). These findings open a new avenue in host-*M. tuberculosis* relevant metabolism, revealing the use of organic nitrogen sources by *M. tuberculosis* during infection.

Although genomic data can be used to construct plausible sets of reactions that might form the core nitrogen metabolic network in *M. tuberculosis* (such as the one shown in *Figure 1*), these models are to some extent incomplete and inaccurate. Utilization of host amino acids as nitrogen sources requires some distinct characteristics from the *M. tuberculosis* metabolic network, which have neither been described or formally investigated to date in model organisms and hence cannot be modelled based on such systems. Another important metabolic network property to be considered is the ability to co-metabolise multiple nutrient (nitrogen) sources.

In contrast to $NH_4^+$, amino acids have been largely under-studied as nitrogen sources for *M. tuberculosis.* In spite of this, there is now overwhelming evidence on the importance of amino acids during infection, highlighted by the profound infection attenuation observed with genetic knockout strains (*Berney et al., 2015*; *Hondalus et al., 2000*). We therefore decided to explore the structure and operation of the *M. tuberculosis* core nitrogen metabolic network with amino acids as nitrogen sources, employing a combination of bacterial physiology, metabolomics and stable isotope labelling experiments.

## Results

### *M. tuberculosis* can take up all proteinogenic amino acids.

As a first investigation on amino acid uptake and utilization by *M. tuberculosis*, we transferred bacteria-laden filters after 5 days' growth on 7H10 media to individual fresh 7H10 agar plates containing 1 mM of each of the 20 proteinogenic amino acids. Cells were harvested 17 hr after transfer. Metabolites were extracted, separated, identified and quantified by liquid-chromatography high-resolution mass spectrometry, following procedures described elsewhere (*de Carvalho et al., 2010*; *Larrouy-Maumus et al., 2013*). The majority of amino acid intracellular pool sizes vary only modestly when *M. tuberculosis* is grown in media containing sole nitrogen sources different to $NH_4Cl$ (*Figure 2a,b*). However, an increase in intracellular pool size is observed for Gly, Ala, Val, Ile, Met, Pro, Phe, Tyr, Trp, Ser, Thr, Arg, and His when *M. tuberculosis* was cultured with the cognate amino acid as sole nitrogen source (highlighted in the diagonal of *Figure 2a*). Importantly, all amino acids present as sole nitrogen source alter the pool size of the cognate amino acid and/or other amino acid in *M. tuberculosis*, demonstrating that they are taken up. On *Figure 2b*, the data from *Figure 2a* are replotted to illustrate individual amino acids changes obtained with *M. tuberculosis* grown with different amino acids as sole nitrogen sources. These data show the final concentrations in samples, not fold-change, compared to $NH_4^+$ conditions. With few exceptions (Met and Trp) no change is observed in the summed amino acid pool size, when *M. tuberculosis* is incubated with different amino acids as the sole nitrogen source. *Figure 2c* contains the data from *Figure 2b* replotted as summed fold-change versus $NH_4Cl$. Trp and His as sole nitrogen sources display significant effects on the summed abundance, but most of the other amino acids do not significantly affect overall pool size. In other words, Trp and His are readily taken up by *M. tuberculosis* and stored at high concentrations. *Figure 2d* shows data for all amino acids, independent of nitrogen source. It is apparent

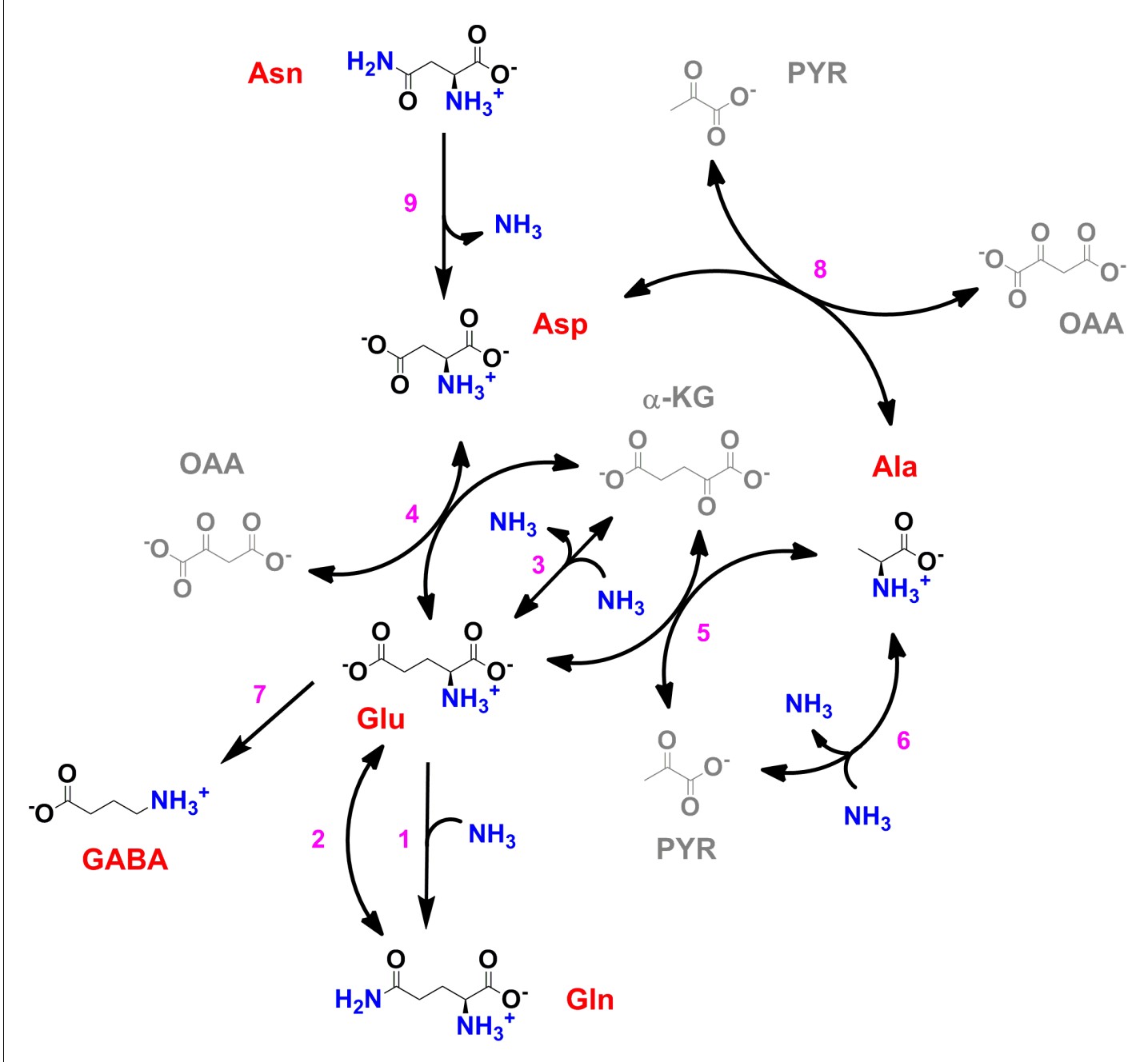

**Figure 1.** Scheme of the core nitrogen metabolic network of *M. tuberculosis*. 1 – Glutamine synthetase (*glnA*1); 2 – glutamate synthase (*gltBD*); 3 – glutamate dehydrogenase (*gdh*); 4 – glutamate/oxaloacetate transaminase (*aspB*); 5 – glutamate/pyruvate transaminase (*aspC*); 6 – alanine dehydrogenase (*ald*); 7 – glutamate decarboxylase (*gadB*); 8 – aspartate/pyruvate transaminase (*aspC*); 9 – asparaginase (*ansA*). Scheme was constructed with data from the Kyoto Encyclopedia for Genes and Genomes (https://www.genome.jp/kegg/kegg2.html) and Mycobrowser (https://mycobrowser.epfl.ch/), and manually curated.

DOI: https://doi.org/10.7554/eLife.41129.003

The following figure supplements are available for figure 1:

**Figure supplement 1.** Common bacterial transcriptional regulators involved in nitrogen metabolism are not present in *M. tuberculosis*.

DOI: https://doi.org/10.7554/eLife.41129.004

**Figure supplement 2.** Different growth kinetics displayed by *M. tuberculosis* (orange circles) and *M. smegmatis* (purple circles) where $NH_4^+$ is the sole nitrogen source.

DOI: https://doi.org/10.7554/eLife.41129.005

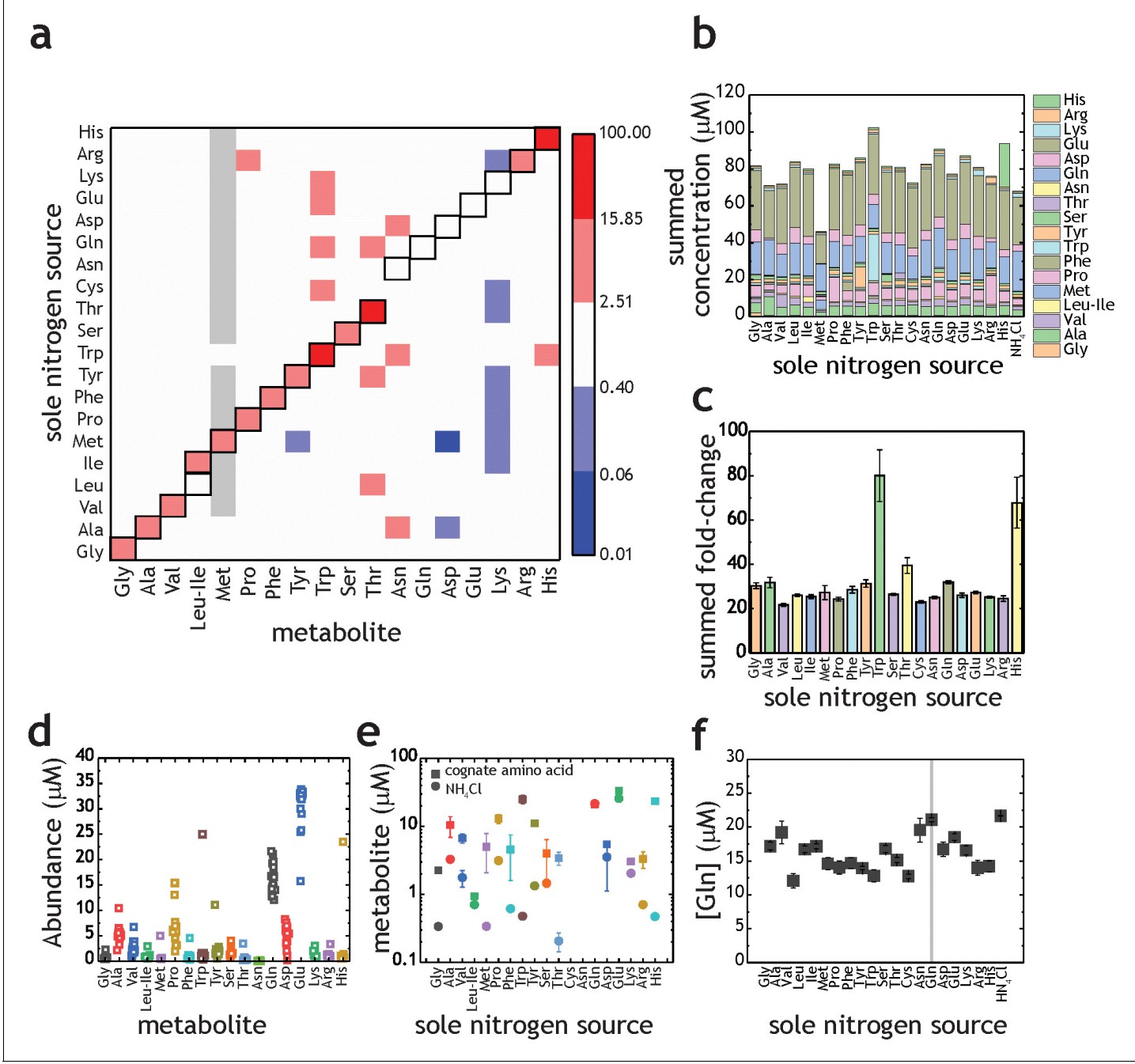

**Figure 2.** Proteinogenic amino acids as sole nitrogen source for *M. tuberculosis*. (**a**) Heatmap illustrating the changes in amino acids (*X*-axis) when *M. tuberculosis* is grown on each individual amino acid as sole nitrogen source (*Y*-axis). Data shown as fold-change (amino acid/$NH_4^+$). Grey squares indicate that abundance of a particular metabolite was too low to be quantified. Cysteine was undetectable in all conditions and was omitted from this plot. Panels (**b–f**) are re-plots of the data shown in panel (**a**). (**b**) Summed abundance of amino acids in each amino acid as sole nitrogen source. (**c**) Data from panel (**b**) presented as fold-change over $NH_4^+$. (**d**) Amino acid abundances irrespective of the sole nitrogen source used, highlighting the variation on each amino acid in different nitrogen sources (e.g. higher variation in Pro and lower in Asn). Each symbol represents the average concentration obtained with a single individual nitrogen source. (**e**) Amino acid concentrations in $NH_4^+$ and in medium containing the cognate amino acid as sole nitrogen source. (**f**) Concentration of Gln in extracts from *M. tuberculosis* grown on different amino acids as sole nitrogen source. All concentrations are final concentrations in lysates obtained from approximately $10^9$ cells, and not concentrations per cell. Data is the average of three biological replicates and representative of two independent experiments.

DOI: https://doi.org/10.7554/eLife.41129.006

that most amino acid concentrations are not significantly altered in different nitrogen sources. Also, it is noteworthy that some amino acid concentrations vary considerably in different nitrogen sources, (e.g. Pro, Asp, Gln, Glu and Ala). *Figure 2e* illustrates amino acid levels observed when the cognate amino acid or $NH_4Cl$ were used as sole nitrogen source. Data further illustrate that nearly all amino acid pool sizes are altered when the cognate amino acid is present in the growth medium, as sole nitrogen source. This data is also highlighted on the diagonal in *Figure 2a*. Curiously, no change is observed in Leu, Asn, Gln, Asp, Glu and Lys when the respective cognate amino acid was added to the growth medium. *Figure 2f* illustrates the concentrations of Gln in *M. tuberculosis* when different amino acids or $NH_4Cl$ are present in the growth medium. Interestingly, very little change in Gln pool size is observed. This is surprising, as Gln is thought to be one of the key indicators of nitrogen levels in cells, alongside with $\alpha$-ketoglutarate (*Senior, 1975*).

Overall, these results indicate that *M. tuberculosis* does not control the pool sizes of all 20 amino acids homeostatically, given that the intracellular concentrations of certain amino acids rise or fall depending on extracellular amino acid/nitrogen source availability. This finding suggests that there are two groups of proteinogenic amino acids in *M. tuberculosis*, the ones that are homeostatically controlled and the ones which intracellular concentration will be affected by extracellular availability. These results also document directly, for the first time, the ability of *M. tuberculosis* to uptake all 20 proteinogenic amino acids.

## Amino acids are superior nitrogen sources, compared to $NH_4^+$

Before carrying out an in-depth analysis of nitrogen metabolism we investigated whether or not the medium used to culture *M. tuberculosis* prior to switching to media with defined sole nitrogen sources could lead to false results. Pre-culture medium composition has been shown to affect carbon metabolism (*de Carvalho et al., 2010*). We 'pre-cultured' *M. tuberculosis* in either standard Middlebrook 7H9 broth (containing Glu and $NH_4^+$) or a 7H9$^{NH4+}$ broth (a synthetic version of Middlebrook 7H9 broth, with $NH_4^+$ as sole nitrogen source), prior to the experiment in 7H9$^{NH4+}$ broth. When pre-conditioned in standard 7H9, growth of *M. tuberculosis* in 7H9$^{NH4+}$ led to a significantly higher biomass accumulation than when pre-conditioned in 7H9$^{NH4+}$*Figure 3—figure supplement 1*. Therefore, without pre-adaptation in the nitrogen source that will be tested, such experiments will likely always overestimate the potential of nitrogen sources. This is particularly problematic when working with compounds that are not able to serve as sole nitrogen source, or are poor sole nitrogen source. Based on these results, all experiments were carried out with cultures that were pre-adapted in a medium of identical composition to the test medium for at least 3 days (unless otherwise stated). Of note, a potential confounding factor in our experiments is the presence of carbon-containing nitrogen sources, such as amino acids, and carbon-free, such as $NH_4Cl$, which could lead to faster grown. However, our media contains an excess of carbon sources (2 g/L glucose and 2.52 g/L glycerol), and therefore it is unlikely that additional carbon in the form of amino acids can account for the results described below.

*Figure 3d* shows representative growth curves obtained in Glu, Gln, Asp, Asn and $NH_4Cl$, as sole nitrogen sources. All four amino acids were superior nitrogen sources to $NH_4Cl$, at all concentrations tested (*Figure 3b and c*), both in terms of doubling rate and final biomass generated. It is noteworthy that pre-adaptation in medium with $NH_4Cl$ as sole nitrogen source shows that *M. tuberculosis* can only optimally utilise $NH_4^+$ as sole nitrogen source until up to 0.25 g/L (4.67 mM). Higher concentrations of $NH_4Cl$ lead to less growth, indicating that high concentrations of ammonium are toxic. Based on these results the following order represents the preferential utilisation of sole nitrogen sources (considering final biomass): Glu > Asp > Asn > Gln > $NH_4^+$. Interestingly, when pre-adapted cultures were grown in medium containing no nitrogen source, growth persisted in all cases (*Figure 3d*). This limited growth is likely due to the low levels of ferric ammonium citrate (0.04 g/L) added to the medium as an iron source. Moreover, this growth was different, depending on nitrogen source: cells pre-adapted to Asp, Asn and $NH_4Cl$ allowed growth to an OD ~1.0, while those pre-adapted to Glu and Gln grew to OD ~0.2. To confirm the 'metabolic conditioning effect' induced by the pre-adaptation medium, we sub-cultured cells after 15 days into fresh medium (with only ferric ammonium citrate). Pre-adaptation with Glu and Gln again led to poor growth, while cells derived from medium containing Asp, Asn and $NH_4Cl$ grew to an OD ~1, in a concentration-dependent manner (*Figure 3e*). In similar experiments, where ferric citrate was used instead of ferric ammonium citrate, negligible growth was observed in derived from media containing Glu and Asp, while slightly

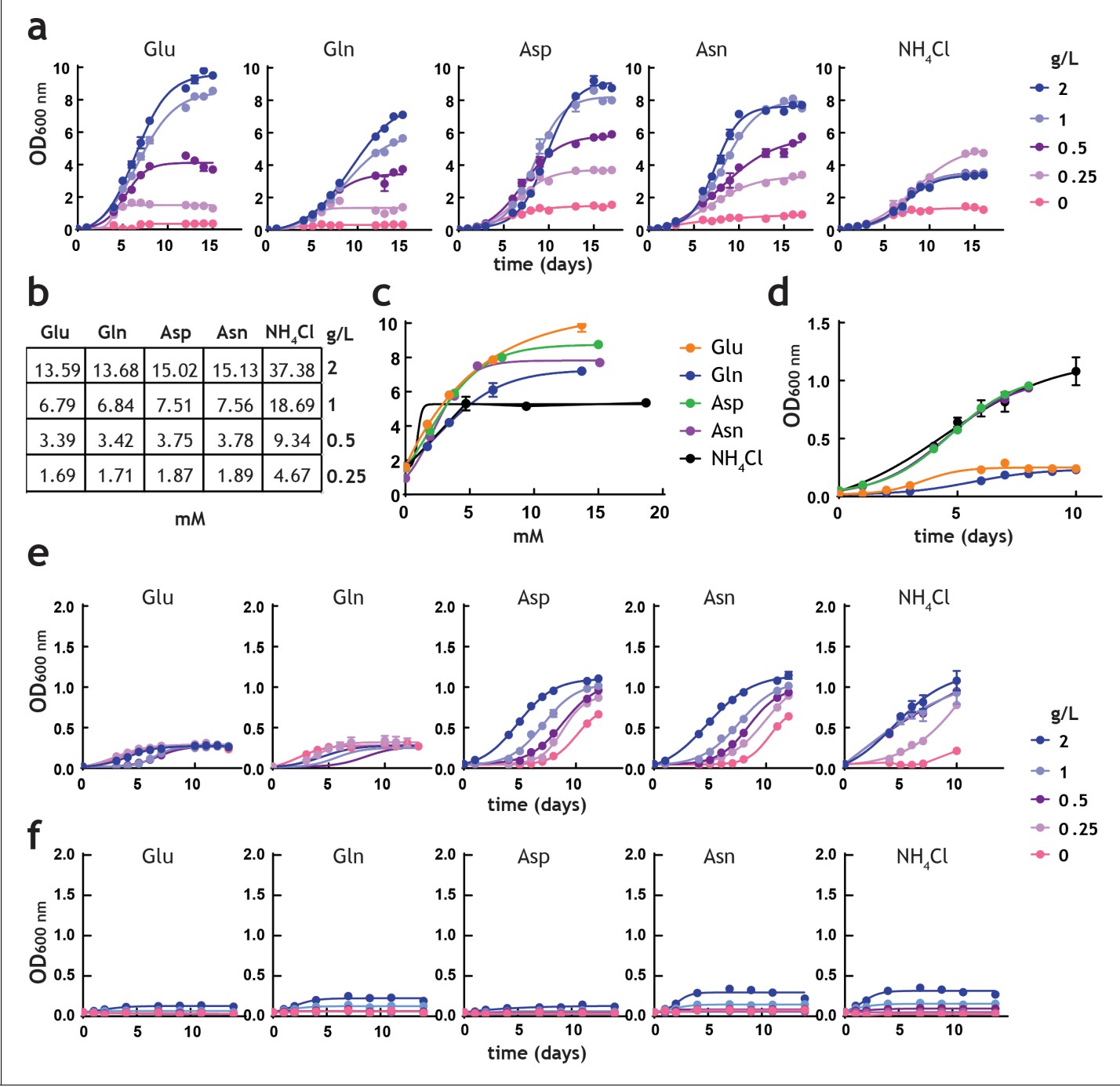

**Figure 3.** Analysis of *M. tuberculosis* growth in pre-adapted nitrogen cultures. (a) Growth curves in 7H9Nx broth (sole nitrogen source). (b) Table with g/L to mM conversions for each nitrogen source used. (c) Replot of final biomass achieved (OD600 nm) for each nitrogen source, after 15 days (a). Solid lines are the fit to a hyperbolic equation, describing saturation. (d) Re-plot of data at no-nitrogen from (a), illustrating different residual growth. (e) Growth curves in 7H9Nx broth without added nitrogen, after cultures were grown for 15 days on nitrogen media (a). 7H9Nx broth still contains low level of nitrogen, in the form of ferric ammonium citrate. (f) Growth curves in synthetic 7H9Nx# broth, lacking nitrogen (ferric ammonium citrate was substituted by ferric citrate), after cultures were grown for 15 days on nitrogen media (a). Symbols are data and solid lines in growth curves are the fit to a sigmoidal equation describing bacterial growth. Data are representative of two independent experiments. Error bars are standard error of the mean.

DOI: https://doi.org/10.7554/eLife.41129.007

The following figure supplement is available for figure 3:

**Figure supplement 1.** Effect of pre-adaptation on sole nitrogen source, prior to growth analysis.
DOI: https://doi.org/10.7554/eLife.41129.008

better growth was observed with cells derived from media containing Gln, Asn and $NH_4^+$ (*Figure 3f*). These results indicate that *M. tuberculosis* does not store nitrogen to any major extent.

Taken together, these results reveal that the amino acids Glu, Gln, Asp, and Asn are superior to $NH_4^+$ as sole nitrogen sources for *M. tuberculosis*, leading to high biomass and faster growth. When traces of $NH_4^+$ are present but no added nitrogen sources have been included, prior sole nitrogen exposure does have an effect on growth, likely indicative of complex metabolism which probably also involves carbon metabolism.

## Utilisation of position-specific nitrogen atoms by *M. tuberculosis*

An essential step in the analysis of nitrogen metabolism with nitrogen sources containing more than one nitrogen atom, such as Gln and Asn, is to define which nitrogen atom(s) is/are being utilised. This characteristic is likely variable and species-specific, and the precise nature of which is currently unclear in *M. tuberculosis*. To confirm which metabolic reactions are likely taking place, we performed labeling experiments with position-specific labelled Gln and Asn (*Figure 4a*). The most direct chemical reactions producing five key amino acids (Glu, Gln, Asp, Asn and Ala) and the label incorporation data obtained from doubly and position-specific labelled $^{15}$N-Gln and $^{15}$N-Asn are shown in *Figure 4b–f*. These results indicate that both nitrogen atoms from Gln and Asn are utilised by *M. tuberculosis* and, specifically that: (i) glutamate synthase is converting the $\delta^{15}$N from Gln into $\alpha^{15}$N-Glu (*Figure 4b*), explaining the incorporation of $\delta^{15}$N from Gln into the $\alpha^{15}$N-Asp, via direct transamination from $\alpha^{15}$N-Glu (*Figure 4d*); (ii) direct transamination between $\alpha^{15}$N-Glu, the other product of the glutamate synthase reaction, and $\alpha^{15}$N-Asp is clearly observed (*Figure 4d*); (iii) when position-specific labelled Asn is used, the dominant form of Asp observed in $\alpha^{15}$N-Asp (*Figure 4d*), indicating that the $NH_4^+$ released by asparaginase is likely assimilated to Gln which is distributed broadly in metabolism (*Figure 4b, c and f*), but only modestly to Asp (*Figure 4d*); (iv) labelled Asn is only detectable when Asn is the nitrogen source (*Figure 4e*), confirming that no Asn synthesis is taking place in *M. tuberculosis*. In contrast to common knowledge, a number of bacteria, including mycobacteria, do not synthesize Asn using asparagine synthetase, but instead they employ a pathway that relies on the amidation of Asp-tRNA$^{Asn}$ (*Javid et al., 2014*) (v) use of either position-specific labelled Gln or Asn, leads to identical labelling of Glu (*Figure 4b*), consistent with access of both $\alpha$ and $\gamma/\delta$ nitrogen atoms; (vi) labelling of Gln with position specific Gln and Asn is indistinguishable, demonstrating that all nitrogen derived from Asn is mobilised through Gln (*Figure 4c*); and (vii) labelling patterns obtained for Ala in the presence of position-specific labelled Gln and Asn are very similar, indicating again that most of the nitrogen derived from Asn is assimilated first into Gln, and then distributed to other metabolites, reflecting the data shown in *Figure 4c*.

## Kinetics of nitrogen metabolism in *M. tuberculosis*

Label incorporation from $^{15}$N Glu, $^{15}$N$_2$-Gln, $^{15}$N-Asp, $^{15}$N$_2$-Asn and $^{15}$NH$_4$Cl obtained under metabolic steady-state, over the course of 17 hr, revealed several important features of *M. tuberculosis* nitrogen metabolism, including different kinetics of $^{15}$N labelling (*Figure 4g and h*, and *Figure 4—figure supplement 1*). As expected, regardless of the nitrogen source, robust label incorporation into amino acids belonging to core nitrogen metabolism was observed, with exception of Asn, which was only observed when cells grew in Asn as sole nitrogen source (*Figure 4g*). It is noteworthy that the Ala pool size and labelling was significantly higher when NH$_4$Cl or Asn was the sole nitrogen source. Also, in agreement with data shown in *Figure 2a*, external amino acid availability does not necessarily correlate with increased intracellular pool size. For example, Glu and Gln are more abundant with Asp and Asn as the sole nitrogen source, respectively (rather than in the cognate amino acid as sole nitrogen source).

Taking the position-specific labelling data and corresponding likely metabolic paths, in conjunction with current biochemical and genetic knowledge of the enzymes of the core nitrogen metabolic network (summarised in *Figure 1*), we calculated exponential labelling rates ($R$) and maximum labelling levels ($L_{max}$) for various core amino acids when Asp, Asn, Glu, Gln and NH$_4$Cl were used as sole nitrogen sources (*Figure 4h* and *Figure 4—figure supplement 1*). $L_{max}$ for different sole nitrogen sources appears to be similar (*Figure 4h*), indicating that, in principle, nitrogen derived from Glu, Gln, Asp, Asn and $NH_4^+$ can reach similar high levels (close to 100%) in the core nitrogen metabolites of *M. tuberculosis* before the first division (17 hr). (*Figure 4h*). In contrast, $R$ values varied

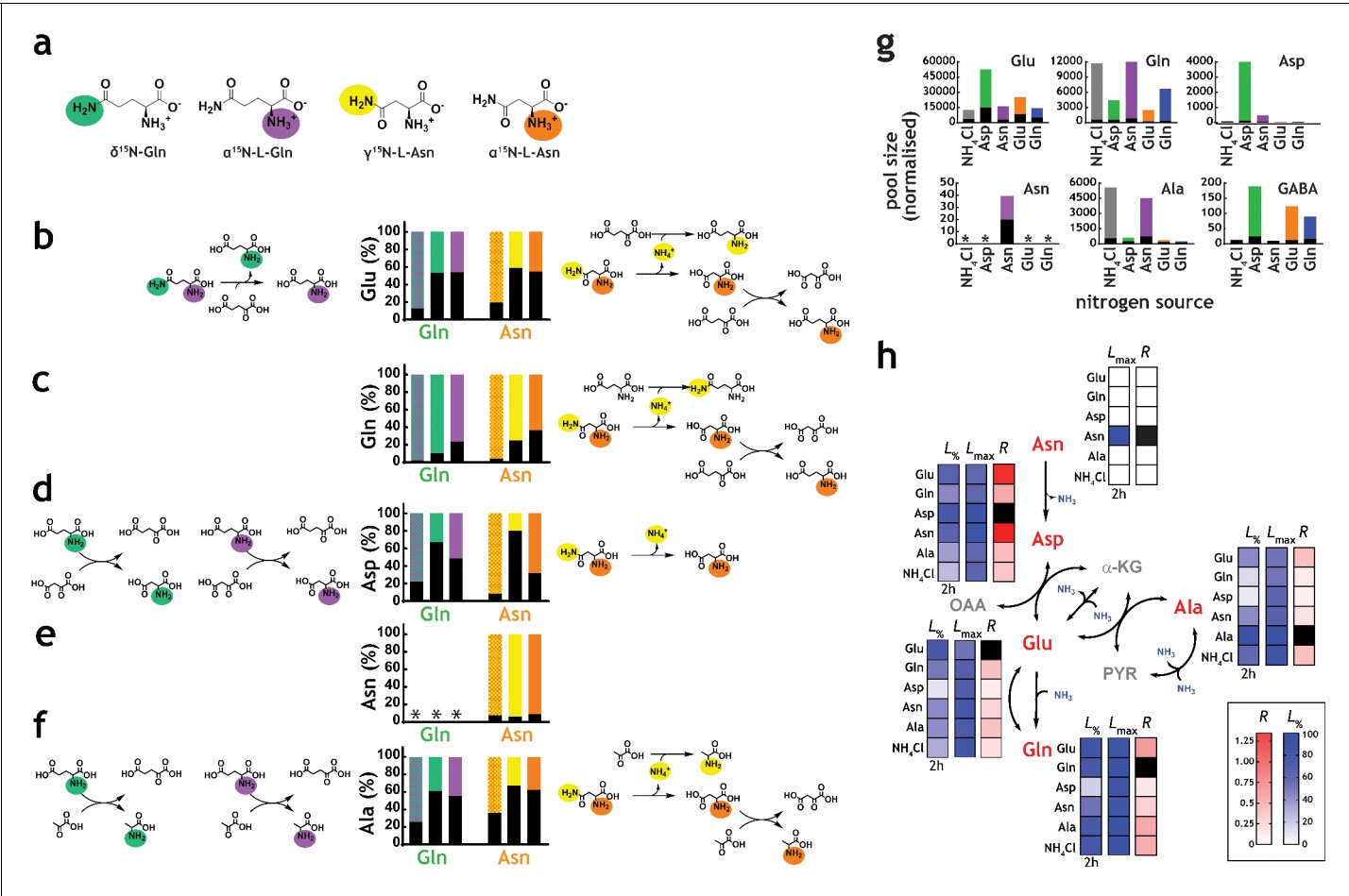

**Figure 4.** Network structure and kinetic analysis of nitrogen utilisation by *M. tuberculosis*. (a) Scheme illustrating the structure and position-specific labelling of nitrogen atoms on Gln and Asn. The following m/z values were used in positive mode (M+H)$^{+}$: Glu – 148.0604, $^{15}$N-Glu – 149.0575, Gln – 147.0764, $^{15}$N-Gln – 148.0735, $^{15}$N$_2$-Gln – 149.0705, Asp – 134.0448, $^{15}$N-Asp – 135.0418, Asn – 133.0608, $^{15}$N-Asn – 135.0578, $^{15}$N$_2$Asn – 135.0548, Ala – 90.0550, and 15N-Ala – 91.0520. (b–f) Data on universally or position-specific labelled Gln or Asn and simplest metabolic routes that would lead to the expected labelling patterns obtained. (b) Labelling of Glu. Glutamate synthase (with Gln) and asparaginase, glutamate dehydrogenase and glutamate/ oxaloacetate transaminase (with Asn). (c) Labelling of Gln. Asparaginase, glutamate dehydrogenase, glutamine synthetase (not shown) and glutamate/ oxaloacetate transaminase, followed by glutamine synthetase (not shown). (d) Labelling of Asp. Glutamate synthase (not shown) and glutamate/ oxaloacetate transaminase, with Gln. Asparaginase is responsible for most of the labelling in Asp, when Asn is the sole nitrogen source. (e) Labelling of Asn. No Asn can be measured in Gln as sole nitrogen source. And most Asn is labelled when Asn is the sole nitrogen source. (f) Labelling of Ala. Glutamate synthase (not shown), glutamate/pyruvate transaminase, with Gln as sole nitrogen source. Asparaginase, alanine dehydrogenase and aspartate/pyruvate transaminase, with Asn as sole nitrogen source. (g) Representative labelling (coloured segment of the bars) and pool sizes for different amino acids obtained after 17 h culture in $^{15}$N-labelled nitrogen sources. Labelling data is coloured by nitrogen source and represents the sum of all labelled species for each ion. (h) Data illustrating quantitative analysis of nitrogen labelling in *M. tuberculosis* in sole nitrogen sources obtained during the course of 17 h. Labelling data (shown in *Figure 4—figure supplement 1*) was fitted to a single exponential rise to a maximum ($L = Lmax \times (1 - e^{-Rt})$). Black squares indicate uptake (cognate amino acid) and not metabolic labelling. Data shown is representative of two independent experiments.

DOI: https://doi.org/10.7554/eLife.41129.009

The following figure supplements are available for figure 4:

**Figure supplement 1.** Kinetics of $^{15}$N label incorporation into core nitrogen metabolites was analysed by high-resolution mass spectrometry.
DOI: https://doi.org/10.7554/eLife.41129.010

**Figure supplement 2.** Minimal perturbations of carbon metabolism accompany utilisation of diverse nitrogen sources by *M. tuberculosis*.
DOI: https://doi.org/10.7554/eLife.41129.011

considerably, depending on the sole nitrogen source present and reactions needed to transfer the $^{15}$N atom to individual metabolites (*Figure 4h*). Once again, it is clear that $NH_4^+$ is not the most efficient nitrogen source for *M. tuberculosis*, as it leads to only modest labelling of key core metabolites, compared to other sole nitrogen sources. $L_{max}$ and $R$ for Asn are only consistently observed when Asn is used as sole nitrogen source, supporting the idea that *M. tuberculosis* has a very small Asn pool (*Figure 4g and h*). $L_\%$ in *Figure 4h* illustrates early (2 hr incubation) labelling of metabolites, and further highlights the differences in $R$ values for each nitrogen source. Of note, the metabolic patterns observed with different nitrogen sources are not secondary, i.e. derived from effects on carbon metabolism. As can be seen in *Figure 4—figure supplement 2*, very little changes in the pool sizes of key metabolites from glycolysis and Krebs cycle are observed and those do not correlate with nitrogen source usage.

## Co-catabolism of amino acids does not improve growth

*M. tuberculosis* has been shown to be able to co-catabolise carbon sources (*de Carvalho et al., 2010*), resulting in a better growth than the one observed in individual carbon sources. Co-catabolism of carbon sources is a metabolic feature highly unusual in bacteria, which usually catabolise carbon sources sequentially, displaying biphasic (diauxic) growth kinetics. In the case of host-adapted pathogens, co-catabolism is thought to contribute to improved survival, although no direct knowledge of co-catabolism of nitrogen sources exists at present. To investigate the potential for this in *M. tuberculosis*, we grew cells in media containing the following combinations of nitrogen sources: $^{15}$N-Glu+$^{14}$N-Gln, $^{14}$N-Glu+U$^{15}$N-Gln, $^{15}$N-Asp+$^{14}$N-Asn, or $^{14}$N-Asp+U$^{15}$N-Asn. All nitrogen source combinations lead to robust labelling of Glu, Gln, Asp, Asn (*Figure 5*), indicating that *M. tuberculosis* is indeed able to take up and co-metabolise nitrogen sources. Extracted ion chromatograms (*Figure 5a*) show significant $^{15}$N metabolism in all conditions (i.e. high levels of labelled metabolites in the absence of labelled nitrogen sources). Mass spectral data (*Figure 5b*) show that Gln (initially labelled or unlabelled) has been metabolised extensively, generating all three isotopologues ($^{14}$N$_2$, $^{14}$N$^{15}$N and $^{15}$N$_2$). Mass spectral analysis further confirms that no Asn is being synthesised in *M. tuberculosis*, as no labelled Asn ($^{15}$N$_2$ or $^{15}$N$_1$) is found when $^{15}$N-Asp is used as nitrogen source and no $^{15}$N-Asn is present when $^{15}$N$_2$-Asn is provided as nitrogen source. *Figure 5c* provides average values and errors for labelling of Glu, Gln, Asp, Asn and Ala, in dual nitrogen sources. Interestingly, Ala labelling appears to derive mainly from Asn, rather than Asp. This suggests that Asn is being hydrolysed to Asp and $NH_4^+$, and likely that Ala is either a main entry point for $^{15}NH_4^+$ or it serves as nitrogen storage. This result is in strict agreement with the very fast and extensive label incorporation of Ala when using $^{15}NH_4^+$ or $^{15}$N-Asn as sole nitrogen sources (*Figure 4g and h*). In spite of clear co-metabolism of two different nitrogen sources, no growth advantage (faster doubling time and highest biomass) is observed *Figure 5—figure supplement 1*. Therefore, it seems that while *M. tuberculosis* uses multiple nitrogen sources simultaneously co-metabolism of related nitrogen sources (*i.e.* Glu/Gln and Asp/Asn) does not lead to improved growth (*i.e.* fastest growth and higher biomass achieved).

## Alanine and alanine dehydrogenase as a fundamental node in nitrogen metabolism

Ala pool size and labelling patterns (*Figure 4g and h*) are incompatible with our current understanding of nitrogen metabolism in *M. tuberculosis*. If $NH_4^+$ utilization, either direct or derived from Asn, proceeded through glutamine synthetase or glutamate dehydrogenase, labelling of Glu would always be greater than Ala, which would be produced by transamination of Glu. However, this is not the case. To investigate Ala metabolism in the context of nitrogen assimilation, we first confirmed whether Ala could serve as a nitrogen source. *Figure 6a* shows that *M. tuberculosis* can grow in the presence of Ala as a sole nitrogen source, or in binary combination of Ala with Glu, Gln, Asp, Asn, or $NH_4^+Cl$. These results are consistent with Ala being utilised as a sole nitrogen source and in combination with other nitrogen sources, but without any growth advantage (mirroring the result observed for Glu/Gln and Asp/Asn co-metabolism). qPCR analysis of transcript levels for asparaginase (*ansA*), glutamine synthetase (*glnA1*), glutamate dehydrogenase (*gdh*) and alanine dehydrogenase (*ald*), in sole nitrogen sources was carried out to define if transcriptional programmes are involved in control of nitrogen metabolism, and in particular of alanine dehydrogenase, despite the current lack of

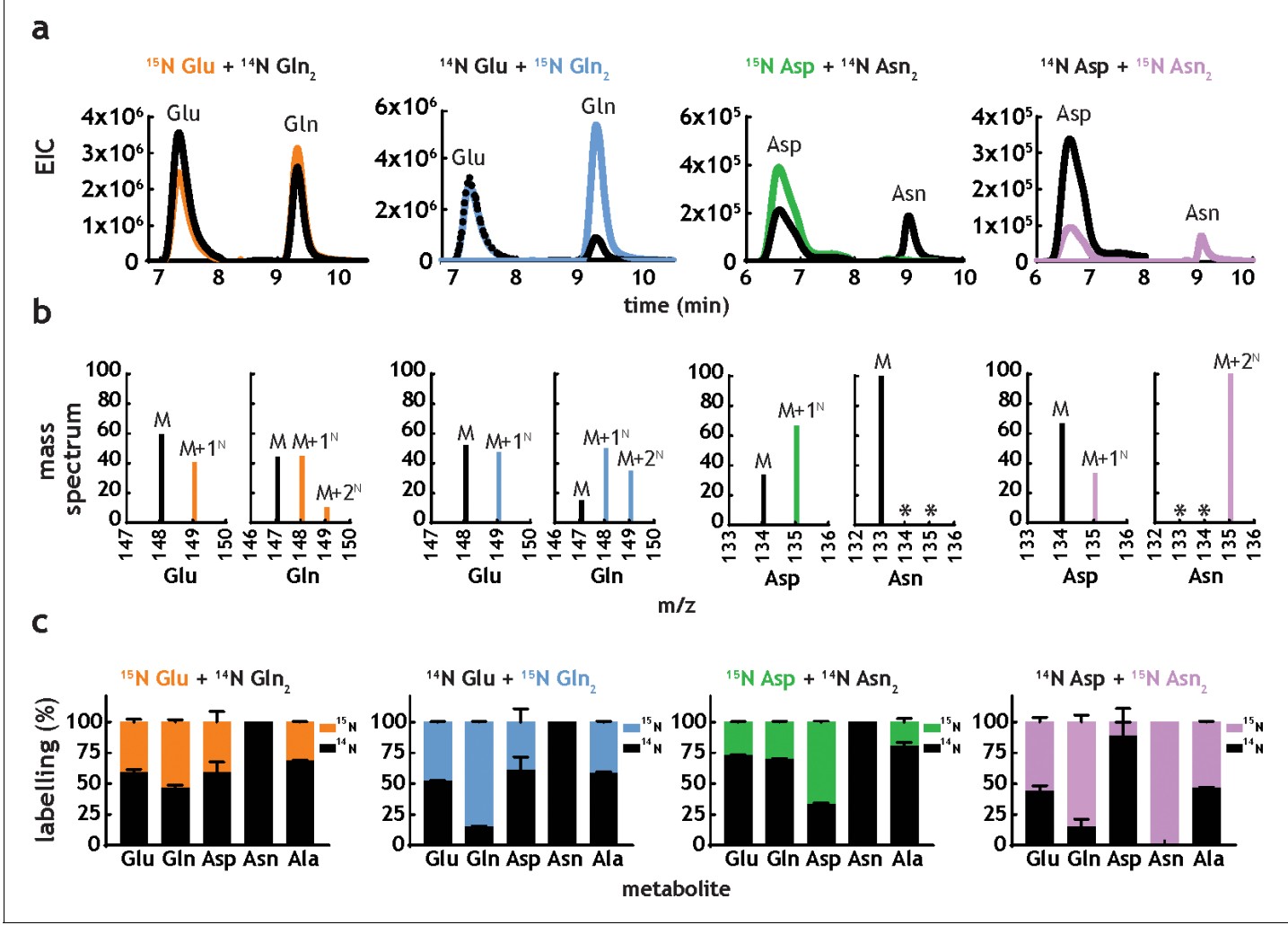

**Figure 5.** *M. tuberculosis* co-metabolises nitrogen sources. (a) Representative extracted ion chromatograms (EICs) for intracellular metabolites from cultures obtained in the presence of dual nitrogen sources (Glu + Gln or Asp +Asn), with one of the nitrogen sources $^{15}$N-labelled. (b) Representative mass spectra corresponding to the metabolites in *Figure 5a*. The following m/z values were used in positive mode $(M + H)^+$: Glu – 148.0604, $^{15}$N-Glu – 149.0575, Gln – 147.0764, $^{15}$N-Gln – 148.0735, $^{15}$N$_2$-Gln – 149.0705, Asp – 134.0448, $^{15}$N-Asp – 135.0418, Asn – 133.0608, $^{15}$N-Asn – 135.0578, and $^{15}$N$_2$Asn – 135.0548. (c) Combined labelling data obtained for the same metabolites, in different combinations of two carbon sources. Bars are averages of three biological replicates, colour indicates labelled metabolites/nitrogen sources and error bars are the standard error of the mean.

DOI: https://doi.org/10.7554/eLife.41129.012

The following figure supplement is available for figure 5:

**Figure supplement 1.** Growth of *M. tuberculosis* in single or dual nitrogen sources.
DOI: https://doi.org/10.7554/eLife.41129.013

potential transcriptional regulators of nitrogen metabolism (*Figure 6b*). Consistent with the hypothesis that alanine dehydrogenase works as a $NH_4^+$ assimilatory route, *ald* RNA levels are found to be higher when *M. tuberculosis* was grown in media with $NH_4^+$, Asn, Asp and Gln, compared to nitrogen-free medium (*Figure 6b*, -N/+N). In addition, *ald* RNA levels are found to be decreased under nitrogen starved conditions, in comparison to *gdh*, *glnA1* and *ansA* RNA levels (*Figure 6b*, 0/-N), suggesting that *ald*-driven nitrogen assimilation is likely more important under nitrogen-rich conditions. The constant levels of expression of *ald* gene when Ala was the sole nitrogen source is likely due to the oxidative deamination of Ala (physiological reaction) is significantly favoured over the reductive amination of pyruvate, and therefore much less enzyme is required.

To define the role of *ald*-encoded alanine dehydrogenase in mobilisation of nitrogen to and from Ala, we compared a *M. tuberculosis* lacking *ald*, to parental and complemented strains (*Giffin et al.,*

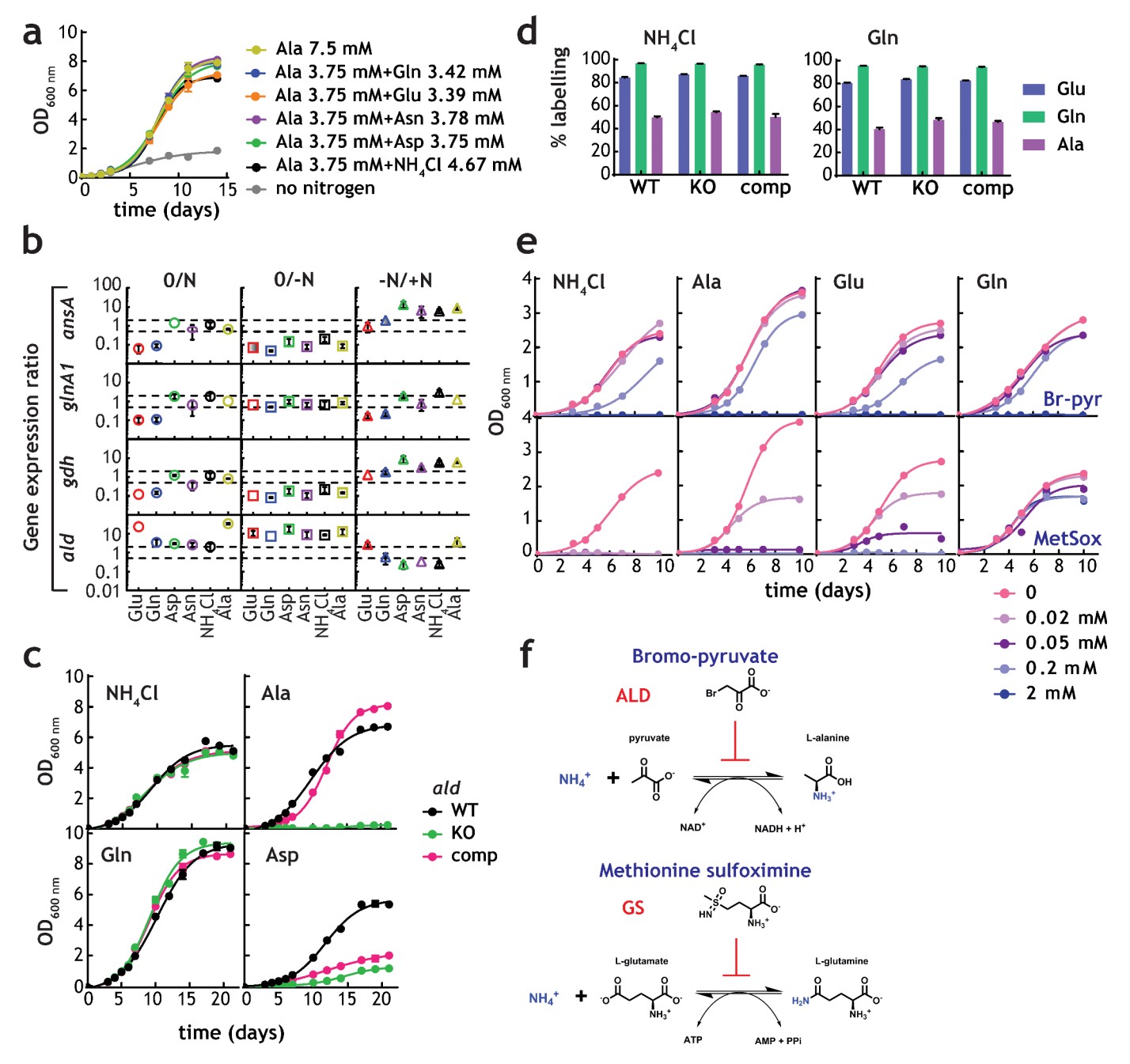

**Figure 6.** Alanine and alanine dehydrogenase roles in *M. tuberculosis* nitrogen metabolism. (**a**) Growth of *M. tuberculosis* on alanine as sole nitrogen source or in combination with a second nitrogen source. (**b**) Gene expression ratios (qPCR) in different nitrogen sources confirms induction of *ald* gene in the presence of $NH_4^+$-containing medium (lower values in the –N/+N plots). "0" indicates the original condition, "-N" indicates medium without a nitrogen source and "+N" indicates medium with a sole nitrogen source.*gdh*, *glnA*1, and *ansA* genes, encoding glutamate dehydrogenase, glutamine synthetase and asparaginase were used as controls, respectively. SigE (Rv1221) was used as internal standard. Symbol colour represents nitrogen source used. Dashed gray lines are used to indicate fold change. Error bars represent standard deviations from three biological replicates. (**c**) Growth of *M. tuberculosis* (WT), *ald* KO, and complemented strains in selected sole nitrogen sources. (**d**) Labelling of selected amino acids obtained with parent, *ald* KO, and complemented strains cultured in $NH_4^+$ or Gln as sole nitrogen sources. (**e**) Growth of *M. tuberculosis* in single nitrogen sources in the presence of various concentrations of bromo-pyruvate (top panels) or methionine sulfoximine (bottom panels), inhibitors of alanine dehydrogenase and glutamine synthetase, respectively. (**f**) Reaction catalysed by alanine dehydrogenase and glutamine synthetase and their inhibitors. Data shown are representative of two independent experiments. Error bars are standard error of the mean.

DOI: https://doi.org/10.7554/eLife.41129.014

*2012*). Genetic disruption of *ald* completely abolished the ability of *M. tuberculosis* to grow when Ala was the sole nitrogen source, with no effect when $NH_4^+$ or Gln were sole nitrogen sources (*Figure 6c*). These results demonstrate that alanine dehydrogenase is essential for assimilation of $NH_4^+$ from Ala, as shown elsewhere (*Giffin et al., 2012*). Interestingly, growth was also significantly diminished in the *ald*-knockout strain when Asp was the sole nitrogen source; however, only partial complementation was obtained (*Figure 6c*). Growth of parent, *ald*-knockout and complemented strains was indistinguishable in $NH_4^+$ as sole nitrogen source, confirming that *ald* is not the main route for $NH_4^+$ assimilation in *M. tuberculosis*. This secondary role of alanine dehydrogenase in assimilation is further supported by lack of changes in label incorporation into Glu, Gln and Ala when parent, *ald*-knockout, and complemented strains were grown with $^{15}NH_4^+$Cl or $^{15}$N-Gln (*Figure 6d*). Final evidence for the essentiality of alanine dehydrogenase in nitrogen assimilation from Ala and secondary role during $^{15}$N assimilation was obtained using the inhibitors of alanine dehydrogenase and glutamine synthetase, bromo-pyruvate (*Bellion and Tan, 1987*) and methionine sulfoximine (*Harth and Horwitz, 1999*), respectively (*Figure 6e and f*). Bromo-pyruvate partially inhibits growth when $NH_4^+$, Ala and Glu are used as sole nitrogen source, but not in Gln. In addition to alanine dehydrogenase, glutamate dehydrogenase is likely also partially inhibited at the concentrations tested, leading to the phenotype observed with bromo-pyruvate in Glu (*Figure 6f*, top panels). Methionine sulfoximine completely abrogated growth in $NH_4^+$ and to a less extent in Ala and Glu but not in Gln as the sole nitrogen source (*Figure 6f*, bottom panels). Together, these results demonstrate that alanine dehydrogenase is essential for utilisation of Ala as sole nitrogen source, but not the main route for $NH_4^+$ assimilation, a task undertaken primarily by glutamine synthetase.

## Discussion

Despite knowing that amino acids can be used by *M. tuberculosis* in vitro for over one hundred years (*Proskauer and Beck, 1894*) only in 2013 and 2014 the first two reports of amino acid utilisation as nitrogen sources by *M. tuberculosis* during infection were published (*Gouzy et al., 2014b*; *Gouzy et al., 2013*). Based on these studies, we set out to evaluate how *M. tuberculosis* utilises nitrogen derived from amino acids in comparison to $NH_4^+$.

An unbiased parallel analysis of amino acid uptake and metabolism reveals for the first time that *M. tuberculosis* can uptake all 20 proteinogenic amino acids. Of interest, it seems that several amino acids can be stored in *M. tuberculosis*, leading to a significant increase in pool size. Such dramatic increase in pool size indicates that the levels of certain amino acids are not homeostatically controlled, but might partially reflect the composition of the medium where bacteria grow. These results coupled with growth kinetic analysis indicates that Glu, Gln, Asp, Asn, Ala are taken up and rapidly metabolised, as sole nitrogen sources, while all the remaining proteinogenic amino acids are taken up by *M. tuberculosis*, but are not utilised as sole nitrogen sources. This accumulation indicates that *M. tuberculosis* metabolic network is adapted to uptake all amino acids, but not necessarily utilise their carbon and nitrogen atoms. For example, *M. tuberculosis* genome lacks genes commonly utilised to degrade histidine, tryptophan, tyrosine and phenylalanine (*Cole et al., 1998*). The α amino group from these amino acids could in theory be transferred and serve as a nitrogen source, however as *M. tuberculosis* cannot catabolise the residual α-keto acid this would likely lead to their intracellular accumulation which in turn would acidify the cytosol, in addition to affect other cellular processes. In contrast, the inability to extract nitrogen from valine, isoleucine and threonine cannot be easily explained, as these amino acids can be degrade or re-routed. Likely the overall architecture of the nitrogen metabolic network, kinetic properties of each of its enzymes, and actual pool size of its key metabolites has a dominant effect on directionality of reactions and actual metabolic flux.

One of the most important results we obtained is that several amino acids are superior nitrogen sources, compared to $NH_4^+$. These results indicate that the vast majority of studies carried out to date on nitrogen metabolism of *M. tuberculosis* using $NH_4^+$ as sole nitrogen source are likely not relevant to physiologic conditions found in the host. More emphasis should be put into microbiological research with amino acids and complex mixtures, as oppose to $NH_4^+$, as sole nitrogen sources. Of note, similar experiments carried out with $NH_4^+$ in *E. coli* (doubling time ~once per 20 min) revealed first-order labelling rates of 14.9, 0.8, >1.6 and 2.8 min$^{-1}$ for Gln, Glu, Ala and Asp, respectively (*Yuan et al., 2006*). Our results with *M. tuberculosis, which doubles every 20 hr*, and $NH_4^+$ are: $9.5 \times 10^{-3}$, $3.7 \times 10^{-3}$, $7.5 \times 10^{-3}$ and $3.5 \times 10^{-3}$ min$^{-1}$ for Gln, Glu, Ala and Asp, respectively.

These labelling rates are approximately 1500-, 216-, >213-, and 800-fold slower than in *E. coli*. Therefore, it seems that *E. coli* takes and utilises nitrogen from $NH_4^+$ between 3- and 26-fold faster, taking into account the doubling rate. Labelling of Glu, Gln, Asp, Asn, Ala and GABA in amino acids as sole nitrogen sources is up to 10-times faster than in $NH_4^+$ as sole nitrogen source, which supports our hypothesis that *M. tuberculosis* has evolved to utilise preferentially host amino acids as nitrogen sources, and to a lesser extent $NH_4^+$.

Nitrogen sources containing more than one nitrogen atom, such as Gln and Asn, were for the first time evaluated with respect to the mobilisation of individual nitrogen atoms. This can only be directly investigated using position-specific labelled Gln and Asn. Our metabolomics results reveal that both nitrogen atoms from Gln and Asn are utilised by *M. tuberculosis*. This result is not easily predicted, as the ability of these to serve as a sole nitrogen sources cannot be deduced from growth kinetics alone. In fact, analysis of position-specific nitrogen utilisation reveals how misleading growth kinetics alone might be as surrogate of uptake/utilization of nutrients. Both Glu and Asp, containing a single nitrogen atom, lead to greater biomass, than Gln and Asn, containing two nitrogen atoms.

Kinetic analysis of nitrogen metabolism employing $^{15}N$-labelled single nitrogen sources (Gln, Glu, Asn, Asp and Ala) reveals that *M. tuberculosis* metabolic network appears to be highly evolved to use very well a number of nitrogen sources, despite *M. tuberculosis* ability to biosynthesise all amino acids. All nitrogen sources tested are able to 'donate' their nitrogen atoms relatively fast, compared to $NH_4^+$.

Co-metabolism of nitrogen sources has never been directly evaluated in any mycobacteria, and only modestly explored in other bacteria. Employing a $^{15}N$-labelled nitrogen source in combination with a $^{14}N$-containing nitrogen source, we demonstrate that at least Glu/Gln and Asp/Asn are simultaneously used by *M. tuberculosis*. As we have suggested for carbon sources (*de Carvalho et al., 2010*), this is a characteristic that is likely important during growth under low nutrient levels, such as when *M. tuberculosis* is found in nutrient-restricted reservoirs intra- or extracellularly. Co-metabolism must lead to a more compartmentalised/optimised used of the metabolic network. Contrary to the results obtained with carbon sources which clearly indicate an advantage of co-metabolism, no growth advantage (doubling-rate and final biomass) is observed with the combined nitrogen sources tested.

Of interest, Ala appears to be labelled faster than Glu, using a number of sole nitrogen sources. This behaviour could not be expected or explained based on our current understanding of *M. tuberculosis* metabolism. Employing *ald* knockout and complemented strains, labelled nitrogen sources, and inhibitors of key nitrogen metabolic enzymes we show that this behaviour is likely due to the reversal of reaction catalysed by alanine dehydrogenase (reductive amination of pyruvate). Our data also reveal that alanine dehydrogenase is absolutely essential for utilisation of Ala as sole nitrogen source, in agreement with the result from *Giffin et al. (2012)*. Given that the Ala pool size dramatically increases during growth in $NH_4^+$ and in Gln, but the enzyme is not essential for growth in $NH_4^+$ and in Gln as sole nitrogen source it is likely that Ala represents a quickly accessible route to store nitrogen in *M. tuberculosis*.

Nitrogen is thought to 'freely' move between molecules once it has been assimilated into Gln and Glu. The latter, is thought to be able to donate its nitrogen atom via a number of transaminases encoded in the genome. Our data does not support such an unregulated and freely reversible exchange. Although transamination reactions are usually fast and have equilibrium constants close to unit, large disparities in the pool sizes of amino acids and related α-keto acids likely impart a single viable direction of nitrogen flow. For example, alanine dehydrogenase is not essential for Ala biosynthesis in *M. tuberculosis*, given that the *ald* knockout strain is not an Ala auxotroph. This supports the existence and functioning of a *bona fide* glutamate/pyruvate transaminase in *M. tuberculosis* and/or the presence of another alanine synthesising pathway/enzyme which can provide enough Ala to support cell wall and protein biosynthesis. Our data unambiguously demonstrates that Ala cannot donate its nitrogen to α-KG, producing Glu and pyruvate. If this is true for other transaminases, it would indicate that Glu is the source of nitrogen for transaminations in *M. tuberculosis*, but only made from glutamate synthase and/or glutamate dehydrogenase, instead of by transamination of amino acids and α-KG. Consequently, amino acids that do not possess alternative enzymatic systems able to remove the α nitrogen, such as dehydrogenases, will likely not serve as nitrogen sources.

Together, these results reveal a number of cellular and molecular details about nitrogen metabolism in *M. tuberculosis* which have escaped detection over the last few decades. Some of these behaviours and characteristics are unique and are likely the results of tens of thousands of years of adaptation to optimise grow and persist in humans as the sole or principal natural reservoir. This extreme ecologic adaptation appears to have shaped *M. tuberculosis* into an 'opportunistic' nutritional generalist. That is, *M. tuberculosis* is able to synthesise every molecule required for its survival but also able to uptake and metabolise a number of different carbon and nitrogen sources from the host.

The ability of *M. tuberculosis* to uptake and utilise a number of proteinogenic amino acids as nitrogen sources likely makes nitrogen uptake an uninviting target for the development of novel antitubercular agents. On the other hand, the ability of *M. tuberculosis* to uptake amino acids which cannot serve as nitrogen sources and might be toxic will likely teach us the true biochemical and metabolic constraints of this pathogen, some of which might find application in drug discovery.

# Materials and methods

**Key resources table**

| Reagent type (species) or resource | Designation | Source or reference | Identifiers | Additional information |
|---|---|---|---|---|
| Strain, strain background (*M. tuberculosis*) | H37Rv | | | MRC-National Insititute for Medical Research |
| Strain, strain background (*M. tuberculosis*) | H37Rv (parent of KO) | doi: 10.1128/ JB.05914–11. | | *Giffin et al., 2012* |
| Strain, strain background (*M. tuberculosis*) | Alanine dehydrogenase KO | doi: 10.1128/ JB.05914–11. | | *Giffin et al., 2012* |
| Strain, strain background (*M. tuberculosis*) | Alanine dehydrogenase complement | doi: 10.1128/ JB.05914–11. | | *Giffin et al., 2012* |
| Software, algorithm | Prism 7 | GraphPad Software | | |
| Software, algorithm | Qualitative Navigator B.07.00 | Agilent software | | |
| Software, algorithm | Profinder B.08.00 | Agilent software | | |
| Chemical compound, drug | Middlebrook 7 H9 | Sigma-Aldrich | M0178 | |
| Chemical compound, drug | ADC supplement | Sigma-Aldrich | M0553 | |
| Chemical compound, drug | OADC supplement | Sigma-Aldrich | M0678 | |
| Chemical compound, drug | Middlebrook 7 H10 | Sigma-Aldrich | M0303 | |
| Chemical compound, drug | Tyloxopol | Sigma-Aldrich | T8761 | |

*Continued on next page*

*Continued*

| Reagent type (species) or resource | Designation | Source or reference | Identifiers | Additional information |
|---|---|---|---|---|
| Chemical compound, drug | Glycerol | Sigma-Aldrich | G5516 | |
| Chemical compound, drug | Sodium sulphate | Sigma-Aldrich | 239313 | |
| Chemical compound, drug | Sodium citrate | Sigma-Aldrich | 51804 | |
| Chemical compound, drug | Pyridoxine | Sigma-Aldrich | P9755 | |
| Chemical compound, drug | Biotin | Sigma-Aldrich | B4501 | |
| Chemical compound, drug | Sodium phosphate dibasic | Sigma-Aldrich | 71642 | |
| Chemical compound, drug | Potassium phosphate monobasic | Sigma-Aldrich | 60220 | |
| Chemical compound, drug | Ferric ammonium citrate | Sigma-Aldrich | F5879 | |
| Chemical compound, drug | Ferric citrate | Sigma-Aldrich | F3388 | |
| Chemical compound, drug | Magnesium sulphate | Sigma-Aldrich | M5921 | |
| Chemical compound, drug | Calcium chloride | Sigma-Aldrich | C8106 | |
| Chemical compound, drug | Zinc sulphate | Sigma-Aldrich | 1724769 | |
| Chemical compound, drug | Copper sulphate | Sigma-Aldrich | C6283 | |
| Chemical compound, drug | Malachite green | Sigma-Aldrich | M9015 | |
| Chemical compound, drug | L-glutamatic acid | Sigma-Aldrich | G1251 | |
| Chemical compound, drug | L-glutamine | Sigma-Aldrich | G3126 | |
| Chemical compound, drug | L-asparagine | Sigma-Aldrich | A4159 | |
| Chemical compound, drug | L-aspartatic acid | Sigma-Aldrich | A9256 | |

*Continued on next page*

*Continued*

| Reagent type (species) or resource | Designation | Source or reference | Identifiers | Additional information |
|---|---|---|---|---|
| Chemical compound, drug | Ammonium chloride | Sigma-Aldrich | A9434 | |
| emical compound, drug | Bromo-pyruvate | Sigma-Aldrich | 16490 | |
| Chemical compound, drug | Methionine sulfoximine | Sigma-Aldrich | M5379 | |
| Chemical compound, drug | L-Alanine-($^{15}$N2) | Cambridge Isotope Laboratory | NLM-454–1 | |
| Chemical compound, drug | L-Asparagine-($^{15}$N2) | Cambridge Isotope Laboratory | NLM-3286 | |
| Chemical compound, drug | L-Asparagine-(amine-$^{15}$N) | Sigma-Aldrich | 489964 | |
| Chemical compound, drug | L-Asparagine-(amide-$^{15}$N) | Cambridge Isotope Laboratory | NLM-120 | |
| Chemical compound, drug | L-Aspartate-($^{15}$N) | Sigma/ Cambridge Isotope Laboratory | 332135/ NLM-718 | |
| Chemical compound, drug | L-Glutamine-($^{15}$N2) | Cambridge Isotope Laboratory | NLM-31328 | |
| Chemical compound, drug | L-Glutamine-(amine-$^{15}$N) | Sigma-Aldrich | 486809 | |
| Chemical compound, drug | L-Glutamine-(amide-$^{15}$N) | Cambridge Isotope Laboratory | NLM-557 | |
| Chemical compound, drug | L-Glutamate-($^{15}$N) | Sigma/ Cambridge Isotope Laboratory | 332143/NLM-135 | |
| Chemical compound, drug | Ammonium chloride -($^{15}$N) | Sigma-Aldrich | 299251 | |
| Chemical compound, drug | Acetonitrile | Fisher | A955-212 | |
| Chemical compound, drug | Methanol | Fisher | A456-212 | |
| Chemical compound, drug | Acetic acid | Fluka | 45740–1 L-F | |
| Sequence-based reagent | Rv0337c-fw | Integrated DNA Technologies | 5'-CACTCCGGTCCACTACCTGT-3' | qPCR primer |
| Sequence-based reagent | Rv0337c-rev | Integrated DNA Technologies | 5'- AGATCGACCATCTGGGTGAG-3' | qPCR primer |

*Continued on next page*

*Continued*

| Reagent type (species) or resource | Designation | Source or reference | Identifiers | Additional information |
|---|---|---|---|---|
| Sequence-based reagent | Rv0858c-fw | Integrated DNA Technologies | 5'- ACGGCACGTACTTCCTATGC-3' | qPCR primer |
| Sequence-based reagent | Rv0858c-rev | Integrated DNA Technologies | 5'- GTTCCACACATCGGCTTGTT-3' | qPCR primer |
| Sequence-based reagent | Rv1178-fw | Integrated DNA Technologies | 5'- ACGAGTGCTACCTGGGATTG-3' | qPCR primer |
| Sequence-based reagent | Rv1178-rev | Integrated DNA Technologies | 5'- AGTAGCTCGGCAACGATCTC-3' | qPCR primer |
| Sequence-based reagent | Rv1538c-fw | Integrated DNA Technologies | 5'- ACTGGAGGGACAATCTCGAC-3' | qPCR primer |
| Sequence-based reagent | Rv1538c-rev | Integrated DNA Technologies | 5'- GAGTGATGACCACCCCATCT-3' | qPCR primer |
| Sequence-based reagent | Rv2220-fw | Integrated DNA Technologies | 5'- GACAAGAGCGTGTTTGACGA-3' | qPCR primer |
| Sequence-based reagent | Rv2220-rev | Integrated DNA Technologies | 5'- GGGTCGTGCACAAAGAAGTT-3' | qPCR primer |
| Sequence-based reagent | Rv2476c-fw | Integrated DNA Technologies | 5'- GTACAGCCTGCTCGACATCA-3' | qPCR primer |
| Sequence-based reagent | Rv2476c-rev | Integrated DNA Technologies | 5'- AGCGCACCGTAAATATCGTC-3' | qPCR primer |
| Sequence-based reagent | Rv2780-fw | Integrated DNA Technologies | 5'- CTTACCACCTGATGCGAACC-3' | qPCR primer |
| Sequence-based reagent | Rv2780-rev | Integrated DNA Technologies | 5'- TAGGCCGATGAGTAGCGAGT-3' | qPCR primer |
| Sequence-based reagent | Rv3565-fw | Integrated DNA Technologies | 5'- TCTACGTGATGGACGTCTGG-3' | qPCR primer |
| Sequence-based reagent | Rv3565-rev | Integrated DNA Technologies | 5'- CACCGAGTATCCCAACTGGT-3' | qPCR primer |
| Sequence-based reagent | Rv1221-fw | Integrated DNA Technologies | 5'- ACCATCACGACCTTGAGTCC-3' | qPCR primer |
| Sequence-based reagent | Rv1221-rev | Integrated DNA Technologies | 5'- AAAGGTCTCCTGGGTCAGGT-3' | qPCR primer |
| Sequence-based reagent | Rv2703-fw | Integrated DNA Technologies | 5'- CCTACGCTACGTGGTGGATT-3' | qPCR primer |
| Sequence-based reagent | Rv2703-rev | Integrated DNA Technologies | 5'- TGGATTTCCAGCACCTTCTC-3' | qPCR primer |

*Continued on next page*

*Continued*

| Reagent type (species) or resource | Designation | Source or reference | Identifiers | Additional information |
|---|---|---|---|---|
| Other | Spin-X centrifuge tube filter cellulose acetate 0.22 mM | Costar | 8160 | |
| Other | Mixed cellulose esters membrane, 0.22 mM | Millipore | GSWP02500 | |
| Other | Acid washed glass beads | Sigma-Aldrich | G1145 | |

## Strains and growth media

*M. tuberculosis* H37Rv was used for growth and metabolic studies. Alanine dehydrogenase Rv2780 knockout, parent (*M. tuberculosis* H37Rv), and complemented strains were generated previously (*Giffin et al., 2012*). Liquid media used for *M. tuberculosis* growth: (a) commercially available Middlebrook 7H9 (Sigma UK) supplemented with (wt/vol) 0.05% tyloxapol, (wt/vol) 0.4% glycerol and albumin-dextrose-catalase (ADC) supplement (Sigma); (b) synthetic 7H9Nx (0.5 g/L sodium sulphate, 0.1 g/L sodium citrate, 1 mg/L pyridoxine hydrochloride, 0.5 mg/L biotin, 2.5 g/L sodium phosphate dibasic, 1.0 g/L monobasic potassium phosphate, 0.04 g/L ferric ammonium citrate, 0.05 g/L magnesium sulphate, 0.5 mg/L calcium chloride, 1 mg/L zinc sulphate, 1 mg/L copper sulphate, 0.4% glycerol, 0.05% tyloxapol, 10% ADC, pH to 6.6) and supplemented with nitrogen source of interest; (c) commercially available Middlebrook 7H10 (Sigma UK) supplemented with 0.5% glycerol and (vol/vol) oleic acid-albumin-dextrose-catalase (OADC) supplement; (d) synthetic 7H10Nx (sodium citrate 0.4 g/L, copper sulfate 1 mg/L, calcium chloride 0.5 mg/L, zinc sulphate 1 mg/L, magnesium sulphate 0.025 g/L, ferric ammonium citrate 0.04 g/L, malachite green 0.25 mg/L, biotin 0.5 mg/L, pyridoxine hydrochloride 1 mg/L, sodium sulfate 0.5 g/L, monopotassium phosphate 1.5 g/L, disodium phosphate 1.5 g/L, Agar 15 g/L, 0.5% glycerol) and supplemented with 10% OADC; and (e) synthetic 7H9Nx[#] (0.5 g/L sodium sulphate, 0.1 g/L sodium citrate, 1 mg/L pyridoxine, 0.5 mg/L biotin, 2.5 g/L sodium phosphate dibasic, 1.0 g/L monobasic potassium phosphate, 0.03 g/L ferric citrate, 0.05 g/L magnesium sulphate, 0.5 mg/L calcium chloride, 1 mg/L zinc sulphate, 1 mg/L copper sulphate, 0.4% glycerol, 0.05% tyloxapol, 10% ADC, pH to 6.6) and supplemented with nitrogen source of interest.

## Metabolite extraction

*M. tuberculosis* was grown in liquid media to mid logarithmic phase and then 1 ml of culture was transferred on 0.22 µm nitrocellulose filter (GSWP02500, Millipore) using vacuum filtration and placed on 7H10Nx agar plates. *M. tuberculosis* loaded filters were then grown at 37˚C for 5 days. On day 5, filters were transferred on chemically identical $^{15}$N 7H10Nx plates for isotopic labelling and metabolites were extracted with acetonitrile/methanol/$_d$H2O 2:2:1 (v/v/v) at −40˚C. Cells were then mechanically disrupted using a Fastprep ryboliser (QBiogene). Samples were centrifuged for 10 min 13,000 rpm at 4˚C, and the supernatant was recovered and filtered through 0.22 µm spin-X centrifuge tube filter (8160, Costar).

## Liquid chromatography-mass spectrometry (LC-MS)

A Cogent Diamond Hydride Type C column (MicroSolv) was used for normal phase chromatography on 1200 LC system (Agilent Technologies) coupled to an Accurate Mass 6220 TOF (Agilent Technologies) mass spectrometer fitted with an MultiMode ion source. Metabolite extracts were mixed with solvent A 1:1 and separated using mobile phase of solvent gradient A and B: 0–2 min, 85% B; 3–5 min, 80% B; 6–7 min, 75%; 8–9 min, 70% B; 10–11.1 min, 50% B; 11.1–14:10 min 20% B; 14:10-18:10 5% B; 18:10–19 85% B. Solvent A was acetonitrile with 0.2% acetic acid and solvent B was ddH2O with 0.2% acetic acid. Reference mass solution (G1969-85001, Agilent Technologies) was used for continuous mass axis calibration. Analytical amino acids standards (Fluka A9906) was used for retention time match. Ions were identified based on their accurate mass, retention time and spectral

information, yielding errors below five ppm. Spectra were analysed using MassHunter Qualitative Analysis B.07.00 and MassHunter Profinder B.08.00 software. Statistical validation of samples/runs were performed using principal component analysis, using Mass Profiler Professional (B.07.01).

## Extraction and analysis of RNA, and qPCR

*M. tuberculosis* was pre-adapted in 7H9Nx medium for 3 days and then grown in identical medium. Cells were harvested at an $OD_{600}$ between 0.8 and 1.0. RNA was extracted using Fast RNA Pro Blue kit according to manufacturer's instructions. DNA was removed by treatment with 3 U RNase-free DNase using the TURBO DNA-free kit (Ambion) according to the manufacturer's instructions and cleaned following RNeasy Mini kit (Qiagen). The concentration of the RNA was determined using a NanoDrop One (Thermo) (Promega) spectrophotometer. Reverse transcriptase PCR was performed using SuperScript IV (Invitrogen), according to the manufacturer's instructions for cDNA synthesis. After cDNA synthesis, qPCR was carried out using the PowerUp SYBR Green Master Mix with ROX (Applied Biosystems) on a QuantStudio 7 Flex Real-Time PCR System. SigE (Rv1221) was used as an internal standard, and the ddCt method was used for the calculation of gene expression ratios. Error bars represent standard deviations from three biological replicates.

## Acknowledgements

We thank Dr James McRae (TFCI) for critical reading of the manuscript. This work was primarily supported by a Wellcome Trust New Investigator Award (104785/B/14/Z) to LPSC. The LPSC lab is also funded by the Francis Crick Institute, which receives its core funding from Cancer Research UK (FC001060), the UK Medical Research Council (FC001060), and the Wellcome Trust (FC001060).

## Additional information

### Funding

| Funder | Grant reference number | Author |
|---|---|---|
| Wellcome | 104785/B/14/Z | Aleksandra Agapova<br>Agnese Serafini<br>Michael Petridis<br>Luiz Pedro Sório de Carvalho |
| Wellcome | Francis Crick Institute Core funding (10060) | Debbie M Hunt<br>Acely Garza-Garcia<br>Luiz Pedro Sório de Carvalho |
| Medical Research Council | Francis Crick Institute Core funding (10060) | Debbie M Hunt<br>Acely Garza-Garcia<br>Luiz Pedro Sório de Carvalho |
| Cancer Research UK | Francis Crick Institute Core funding (10060) | Debbie M Hunt<br>Acely Garza-Garcia<br>Luiz Pedro Sório de Carvalho |

The funders had no role in study design, data collection and interpretation, or the decision to submit the work for publication.

### Author contributions

Aleksandra Agapova, Formal analysis, Investigation, Methodology, Writing—original draft, Writing—review and editing; Agnese Serafini, Investigation, Writing—review and editing; Michael Petridis, Formal analysis, Investigation, Methodology; Debbie M Hunt, Investigation; Acely Garza-Garcia, Resources, Methodology; Charles D Sohaskey, Resources, Writing—original draft; Luiz Pedro Sório de Carvalho, Conceptualization, Resources, Supervision, Funding acquisition, Methodology, Writing—original draft, Project administration, Writing—review and editing

### Author ORCIDs

Acely Garza-Garcia http://orcid.org/0000-0003-0307-0138
Luiz Pedro Sório de Carvalho http://orcid.org/0000-0003-2875-4552

**Decision letter and Author response**
Decision letter https://doi.org/10.7554/eLife.41129.019
Author response https://doi.org/10.7554/eLife.41129.020

## Additional files

### Supplementary files
• Transparent reporting form
DOI: https://doi.org/10.7554/eLife.41129.015

### Data availability
Metabolomics data used on this study are available via Zenodo (DOI 10.5281/zenodo.2551162).

The following dataset was generated:

| Author(s) | Year | Dataset title | Dataset URL | Database and Identifier |
|---|---|---|---|---|
| Agapova A, Serafini A, Petridis M, Hunt DM, Garza-Garcia A, Sohaskey CD, de Carvalho LPS | 2019 | Metabolomics data from Flexible nitrogen utilisation by the metabolic generalist pathogen Mycobacterium tuberculosis | https://zenodo.org/record/2551162#.XFgrZBn7RTY | Zenodo, 10.5281/zenodo.2551162 |

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
