## [Decision Letter]

Thank you for submitting your article "Flexible nitrogen utilisation by the metabolic generalist pathogen *Mycobacterium tuberculosis*" for consideration by *eLife*. Your article has been reviewed by peer reviewers, and the evaluation has been overseen by a Reviewing Editor and Gisela Storz as the Senior Editor. The following individuals involved in review of your submission have agreed to reveal their identity: Martin Voskuil (Reviewer #2).

The reviewers have discussed the reviews with one another and the Reviewing Editor has drafted this decision to help you prepare a revised submission.

Summary:

The submission by de Carvalho and colleagues reports a comprehensive analysis of nitrogen assimilation into amino acids in *M. tuberculosis*. The study demonstrates that *M. tuberculosis* has the ability to import all 20 proteinogenic amino acids and unlike *Escherichia coli, M. tuberculosis* does not retain homeostatic control over amino acid pools. This work further shows that Glu, Gln, Asp, Asn and Ala can serve as nitrogen sources in contrast to other amino acids and are superior nitrogen sources when compared to free ammonium. The enzymatic mechanisms underlying the nitrogen transfer between Glu, Gln, Asp and Asn are demonstrated using stable isotope labelled amino acids and it is further demonstrated that these amino acids can be co-metabolized as nitrogen sources. An intriguing pattern of transfer of nitrogen to alanine is observed, which demonstrates that this amino acid is central in nitrogen metabolism, with alanine dehydrogenase playing a critical role in nitrogen assimilation. The data argue against Glu/Gln being the starting point for downstream free exchange of nitrogen between various amino- and ketometabolites. Instead, there is a very specific directionality with respect to nitrogen transfer. Overall, this is an in-depth thorough study of nitrogen assimilation and incorporation into key metabolites in *M. tuberculosis* showing aspects of nitrogen assimilation and control of biosynthesis of key amino acid that differs significantly from model bacteria.

Key findings:

1) The demonstration that *M. tuberculosis* has the capacity to import all 20 amino acids and that intracellular pools are not homeostatically controlled but rather depend on extracellular abundance.

2) Using preconditioned cultures, Glu, Gln, Asp and Asn emerge as good nitrogen sources to support growth of *M. tuberculosis* when compared to ammonium. Both nitrogen atoms from Gln and Asn are used by *M. tuberculosis*.

3) The demonstration that *M. tuberculosis* can co-catabolize two nitrogen sources however, this does not lead to a growth advantage as has been previously established for carbon co-catabolisation.

4) The authors resolve seemingly paradoxical observation regards Ala metabolism by demonstrating that alanine dehydrogenase is required for metabolism of Ala as a sole nitrogen source.

5) The demonstration of directionality with respect to nitrogen transfer, for example, the data showing that alanine cannot be transaminated directly with ketoglutarate as co-substrate.

Major Comment:

1) One of the most interesting findings from this study is the fact that pre-growth in Asp, Asn and NH_4_Cl allowed for much more growth after removal of nitrogen sources compared to pre-growth with Glu or Gln. Unfortunately, this finding was not pursued. There is a possibility that trace ammonium from the iron source of ferric ammonium citrate, was responsible for this. This experimental design introduces an unnecessary confounder to an otherwise carefully conducted study. A preferable approach would be to use an iron source that does not contain ammonium for this experiment to demonstrate if different nitrogen sources may allow for differences in nitrogen storage. To address this, the authors are requested to repeat the experiment (shown in Figure 3E) where cells were sub-cultured after 15 days into fresh medium with ferric ammonium citrate but replacing the iron source to one without ammonium.

---

## [Author Response]

Major Comment:1) One of the most interesting findings from this study is the fact that pre-growth in Asp, Asn and NH_4_Cl allowed for much more growth after removal of nitrogen sources compared to pre-growth with Glu or Gln. Unfortunately, this finding was not pursued. There is a possibility that trace ammonium from the iron source of ferric ammonium citrate, was responsible for this. This experimental design introduces an unnecessary confounder to an otherwise carefully conducted study. A preferable approach would be to use an iron source that does not contain ammonium for this experiment to demonstrate if different nitrogen sources may allow for differences in nitrogen storage. To address this, the authors are requested to repeat the experiment (shown in Figure 3E) where cells were sub-cultured after 15 days into fresh medium with ferric ammonium citrate but replacing the iron source to one without ammonium.

Thanks for pointing this out. We initially thought that the trace ammonium derived from ferric ammonium citrate would not be a problem. In fact, it is only a problem under “no nitrogen” conditions. We carried out the experiment requested by the reviewers twice, independently, and we show that under strict “no nitrogen” only very slight growth is observed, now shown in Figure 3F. We opted to keep the data shown in Figure 3E, as that is a better mimic of the standard 7H9 broth used by many labs. But clarified in the text that the experiment in Figure 3F is the correct “no nitrogen” one.

The text now reads “In similar experiments, were ammonium citrate was used instead of ferric ammonium citrate, negligible growth is observed in derived from media containing Glu and Asp, while slightly better growth was observed with cells derived from media containing Gln, Asn and NH_4_^+^ (Figure 3F). These results indicate that *M. tuberculosis* does not store nitrogen to any major extent.” And “When traces of NH_4_^+^ are present but no added nitrogen sources have been included, prior sole nitrogen exposure does have an effect on growth, likely indicative of complex metabolism which probably also involves carbon metabolism.*”*